# A Cross-Sectional Study to Assess mRNA-COVID-19 Vaccine Safety among Indian Children (5–17 Years) Living in Saudi Arabia

**DOI:** 10.3390/vaccines11020207

**Published:** 2023-01-17

**Authors:** Marya Ahsan, Riyaz Ahamed Shaik, Ayaz K. Mallick, Saeed S. Banawas, Thamer A. M. Alruwaili, Yousef Abud Alanazi, Hayat Saleh Alzahrani, Ritu Kumar Ahmad, Mohammad Shakil Ahmad, Faisal Holil AlAnazi, Fahad Alfhaid, Mohammed Zaid Aljulifi, Vini Mehta, Abdalah Emad Almhmd, Abdulaziz S. D. Al Daham, Mutlaq M. M. Alruwaili

**Affiliations:** 1Department of Pharmacology, College of Medicine, Imam Mohammad Ibn Saud Islamic University (IMSIU), Riyadh 13317, Saudi Arabia; 2Department of Family & Community Medicine, College of Medicine, Majmaah University, Al Majmaah 11952, Saudi Arabia; 3Department of Clinical Biochemistry, College of Medicine, King Khalid University, Abha 61421, Saudi Arabia; 4Department of Medical Laboratories, College of Applied Medical Science, and Health and Basic Sciences Research Center, Majmaah University, Majmaah 11952, Saudi Arabia; 5Departments of Biomedical Sciences and Microbiology, Oregon State University, Corvallis, OR 97331, USA; 6Department of Pediatrics, College of Medicine, Jouf University, Aljouf 72388, Saudi Arabia; 7Department of Pediatrics, College of Medicine, Majmaah University, Majmaah 11952, Saudi Arabia; 8Department of Clinical Science, College of Medicine, Princess Nourah bint Abdulrahman University, Riyadh 11671, Saudi Arabia; 9Department of Physiotherapy, College of Applied Medical Science, Buraydah Private Colleges, Buraydah 51418, Saudi Arabia; 10Department of Medicine, College of Medicine, Majmaah University, Majmaah 11952, Saudi Arabia; 11Department of Public Health Dentistry, Dr.D.Y.Patil Dental College and Hospital, Dr.D.Y.Patil Vidyapeeth, Pune 411018, India; 12College of Medicine, Majmaah University, Majmaah 11952, Saudi Arabia; 13College of Medicine, Jouf University, Aljouf 72388, Saudi Arabia

**Keywords:** mRNA vaccine, COVID-19, adverse events following immunization (AEFI), adverse events of special interest (AESI), myocarditis

## Abstract

The objective of this study is to assess the frequency and severity of adverse events following immunization (AEFI) in Indian children aged 5–17 years who received the Pfizer-BioNTech mRNA COVID-19 vaccine, as well as to investigate for predictors of AEFI. To examine AEFI following the first and second doses of Pfizer’s vaccine, semi-structured questionnaires were distributed as Google forms at Indian schools in Saudi Arabia. The 385 responses included 48.1% male and 51.9% female children, with 136 responses of children aged 5–11 years (group A) and 249 responses from children aged 12–17 years (group B). Overall, 84.4% of children had two shots. The frequency of AEFI was reported to be higher after the first dose than after the second (OR = 2.12, 95% CI = 1.57–2.86). The reported AEFIs included myalgia, rhinitis, local reaction with fever, a temperature of 102 °F or higher, and mild to moderate injection site reactions. While group B frequently reported multiple AEFIs, group A typically reported just one. Local reaction with low grade fever was more frequently reported in group B after the first dose (24.1%) and second dose (15.4%), while local reaction without low grade fever was most frequently observed in group A after the first (36.8%) and second dose (30%). Only prior COVID-19 infection (OR = 2.98, 95% CI = 1.44–6.2) was associated with AEFI after the second dose in the study sample, whereas male gender (OR = 1.71, 95% CI = 1.13–2.6) and prior COVID-19 infection (OR = 2.95, 95% CI = 1.38–6.3) were predictors of AEFI after the first dose. Non-serious myocarditis was reported by only one child. According to the analysis conducted, the Pfizer’s mRNA COVID-19 vaccination was found to be safe in Indian children.

## 1. Introduction

The World Health Organization (WHO) declared COVID-19, which is caused by the new coronavirus SARS-CoV-2, as the sixth worldwide public health emergency in 2020 [1]. The viruses’ high global infectivity and mortality were caused by their capacity to spread through droplets and endure in aerosols for up to three hours [2]. To de-escalate the transmission of COVID-19, the WHO constantly issued cautions for the general population, advising avoiding social contact, avoiding large gatherings, frequently donning masks, and isolating affected individuals and their close connections. Children experienced a milder form of COVID-19, but multisystemic inflammatory syndrome and long-term post-COVID-19 problems were always a possibility. Additionally, studies have emphasized the detrimental psychological effects of social isolation, including psychological issues, insufficient sleep, and a poor quality of life [3].

WHO approved the much-awaited COVID-19 vaccine in December 2020. As of October 2021, Pfizer-mRNA BioNTech’s COVID-19 vaccine was approved for emergency use in children aged 5 to 11 years [4,5]. Clinical study safety data mainly reported mild to moderate side effects like local responses, headache, fever, fatigue, myalgia, and lymphadenopathy [6]. However, reports of serious adverse events of special interest have emerged (AESI) [7]. Poor vaccination rates among children were driven by worries about the vaccine’s safety and the highly politicized anti-vaccination movement. Also, the lack of scientific data from large scale studies in pediatric age group regarding the safety and adverse effect of COVID-19 vaccines aggravated the hesitancy towards vaccination. Only 25% of parents in the US responded favorably to vaccination, with over 40% outright rejecting to vaccinate their children, according to the Kaiser Family Survey report. A lack of real-time data, adverse effects, and long-term post-vaccination issues were some of the factors contributing to this hesitation [8,9]. Additionally, the immunological response a vaccination induces within ethnic groups can vary, leading to a variety of responses in terms of efficacy and limitations [10]. 

In many countries such as India, there is no data available regarding the safety of mRNA vaccine from Pfizer-BioNTech as it is not accessible in India. However, Pfizer-BioNTech’s covid vaccination has been given to Indians who live abroad. Over 77% of Saudi Arabia’s citizens and residents have received at least one dose of immunization as a result of extensive vaccination campaigns [11]. Saudi Arabia has a sizable Indian population of over 2.5 million. Moreover, there are 39 Indian schools in Saudi Arabia as notified by the Embassy of India, Riyadh, which accounts for a large number of Indians in the pediatric age group. Therefore, we undertook this opportunity to conduct this study to gather real-world data on adverse events following immunization (AEFI) with Pfizer-mRNA BioNTech’s COVID-19 vaccine and assess its safety among Indian children 5–17 years of age residing in Saudi Arabia. 

## 2. Materials and Methods

This study was conducted as a cross-sectional questionnaire-based survey at Imam Muhammad Ibn Saud Islamic University (IMSIU), Riyadh, between April–September 2022. Institutional Review Board approval was granted before commencing the study (Ref.220/2022). After obtaining informed consent, information was collected from parents or guardians.

### 2.1. Sample Size

Keeping the confidence level at 95%, the margin of error (ME) at 5%, and the Standard deviation (SD) at 0.5, the sample size was calculated as 385.

### 2.2. Study Design & Tool

Indian schools in significant Saudi Arabian cities circulated a semi-structured questionnaire to parents via Google forms in order to gather data. An exponential non-discriminative snowball technique was used to further distribute the questionnaire among the Indian population within the Kingdom. The study used information from children of Indian descent who were between the ages of 5 and 17, who were of either gender, and had received one or two doses of the mRNA COVID-19 vaccine from Pfizer-BioNTech. We received 408 responses, of which 23 (5.6%) responses were not included in the study. Incomplete questionnaires, questionnaires with discrepancies, and children that received a COVID-19 vaccination other than the mRNA vaccine from Pfizer-BioNTech were excluded. After applying the exclusion criteria, once the sample size was achieved, the link to the Google form was disabled. 

Written information about COVID-19 vaccinations, AEFI, and its reporting was provided to parents and guardians in easy-to-understand English language along with the questionnaire. In terms of data collection, three sections made up the online survey. Part A of the questionnaire collected demographic data such as age, gender, personal and family history of allergy, comorbidity, history of COVID-19 infection before and after vaccination, history of any unwanted event with routine vaccination, and the number of doses of mRNA vaccine received. Part B and C of the questionnaire collected details to describe the adverse event following immunization (AEFI), which is “any unwanted medical occurrence following immunization and does not necessarily have a causal relationship with the vaccine” [12]. Details of type of unwanted event, its temporal relation with vaccination, duration, need for any intervention, utilization of health resources, and outcome following 1st dose and 2nd dose of vaccine were collected in Part B and C, respectively.

The Hartwig-Seigel scale was used to classify the severity of the adverse event as ‘mild’, ‘moderate’, and ‘severe’ [13]. Any adverse event leading to death, permanent disability, hospitalization, or increase in hospital stay was categorized as “serious” [14]. Details of any AESI as per Brighton-Collaboration’s case definition were also sought [15]. 

### 2.3. Statistical Analysis

For the purpose of statistical analysis, the collected data was divided into two groups: A & B, where A represented younger children (5–11 years old) and B represented adolescents (12–17 years old). The statistical analysis was performed using the Statistical Package for Social Sciences 26.0 (SPSS/PC; SPSS 26.0, Chicago, IL, USA). For continuous data, the mean SD was used, while categorical variables were reported as percentages. The categorical variables were compared using the chi-square test. Age, gender, allergy history, adverse reactions to past routine vaccinations, and prior COVID-19 infection were all evaluated using binary logistic regression as AEFI predictors. An Odds ratio with 95% CI was computed. A p-value of less than 0.5 was used as the threshold for significance. 

## 3. Results

Among the 385 responses collected, 136 included data from children between 5–11 years of age (group A), while 249 were from children between 12–17 years of age (group B). The mean age of the children in our study was 12.25 ± 3.29 years. The majority of the children (55.3%) were residents of Riyadh province of Saudi Arabia, followed by the Eastern province (18.4%). In the study, 48.1% of the children were males, and 51.9% were female. However, in this study, group A (61.8%) contained noticeably more males than group B (40.6%) (p-0.001). Nearly 84.4% of children had received two doses of the vaccine, with 70% of children in group A and 92% of children in group B receiving both doses. (Table 1).

Among the participants, 12.2% had a history of confirmed COVID-19 infection before vaccination, while 3.9% were suspected of having COVID-19 infection. A history of allergies was reported by 8.3% of participants, while 1.3% were unsure. Children in group A had a higher prevalence of allergies (11%) compared to adolescents in group B (*p* < 0.01). In this study, dust was the allergen that was most frequently reported (8.6%). A total of 10.1% of children reported a family member with allergies; this proportion was higher in group A participants (16.2%) (*p* < 0.05). Additionally, 2.9% of children reported having asthma, but only one had eczema and another had adrenal insufficiency. In overall analysis, only 4.4% of children had ever experienced unintended side effects from routine vaccinations. (Table 1).

### 3.1. Relationship with Dose

As seen in Figure 1, 43% of children in study reported AEFI after receiving the second dose of the COVID-19 vaccine, while nearly 62% of children reported AEFI after receiving the first dose (OR = 2.116, 95% CI = 1.57–2.86, *p* < 0.001). In children in group A, AEFI was recorded in 59% of cases after the first dosage and in 41% of cases after the second dose (OR = 2.05, 95% CI = 1.2–3.49). Similar to group A, only 44% of participants in group B reported experiencing an adverse event following the second dose (OR = 2.18, 95% CI = 1.51, 3.43). (Figure 1, Table 2 and Table 3) A Chi-square test was significant between the first and second dose in group A (5–11 years, *p* < 0.01), group B (12–17 years, *p* < 0.001), and in the total number of children (5–17 years, *p* < 0.001).

After receiving the first dose of the vaccine, the study found that local reactions alone, such as pain, swelling, and redness at the injection site (22%), low-grade fever, defined as a fever of less than 102 °F (15.6%), local reaction combined with low-grade fever (18.4%), high-grade fever, defined as a fever of more than 102 °F (1.8%), a local reaction combined with high fever (3.6%), high fever alone (3.6%), and rhinorrhea (1%) were the presentation of AEFI. (Figure 2) After receiving the second dosage of the vaccine, the majority of patients (56.9%) did not report any side effects. After the second dose, local reactions at the injection site alone (15.4%), low grade fever (11.7%), or both local reactions and low-grade fever (11.7%) were frequently observed.(Figure 3).

### 3.2. Relationship with Gender

Study findings also shows a higher percentage of male children (68%) than female children (55.5%) experienced adverse events (AEs) following the first dose of the vaccination (O = 1.71, 95% CI = 1.13–2.5, *p* = 0.011). After the first dose, there was a significant gender difference in group B, with 74% of the male children reporting AEs as opposed to 55% of the female children (*p* < 0.01) (Table 2 and Table 3).

### 3.3. Relationship with Past COVID-19 Infection

A total of 80.9% of children who had previously contracted COVID-19 infection reported side effects after the first dosage (OR = 2.95, 95% CI = 1.38–6.3, *p* = 0.005), whereas 66.7% of children who had previously contracted COVID-19 reported side effects after the second dose (OR = 2.98, 95% CI = 1.44–6.2, *p* = 0.003) (Table 2 and Table 3). Following the first dosage of the vaccine, researchers did not find any correlation between COVID-19 infection and specific AEFIs. After the second dose of the vaccine, study findings show a relationship between prior COVID-19 infection and local injection site reaction (OR = 2.26, 95% CI = 1.12, 4.56, *p* = 0.023). 

### 3.4. Relationship with History of Allergy and AEFI with Previous Vaccines

We did not find any difference in the AEFI frequency rates among those who had allergy or a history of unwanted events with routine vaccination (Table 3).

### 3.5. COVID-19 Infection after Vaccination

In our study, 13.8% of children reported confirmed COVID-19 infection after two doses of the vaccine. There was no difference in the incidence of COVID-19 infection post-vaccination in the two groups (*p* > 0.05). (Table 1).

### 3.6. Adverse Event Characteristics in Group A (5–11 Years)

Study findings showed that pain, redness, and swelling at the injection site were recorded 62 times after the first dose and 33 times after the second dose while examining the adverse events reported in group A. On all counts, the local reaction developed within 1–3 days of vaccination and continued for 1–4 days in the majority of individuals. After the first dose, it continued for 4–7 days in one child, and after the second dose, it persisted for 3 children. After the first and second dosages, in 45/62 children and 24/33 children, respectively, no treatment was offered, and it subsided on its own. On the Hartwig-Seigel scale, their severity was rated as “mild.” After the first and second dosages, respectively, 17/62 and 9/33 children received treatment at home for “moderate” severity cases. 

After the first dose and the second dose, respectively, 29 and 6 cases of low-grade fever were recorded. In comparison, moderate high-grade fever was reported in 1 and 6 cases after the first and second dose, respectively, in group A. Throughout each case, fever started within 1–3 days. However, the patients’ high fever persisted for 4–7 days. After receiving the first dose, low-grade fever was mild in 14 cases and moderate in 15 children, and moderate in all cases after receiving the second dose. In this study, group A patients made a full recovery from the recorded adverse effects (Table 4).

### 3.7. Adverse Event Characteristics in Group B (12–17 Years)

On analyzing the adverse events reported in group B, we found that local reaction at the injection site was the most commonly reported event after the first (*n* = 108) and second dose (*n* = 66). The onset was within 1–3 days and lasted for 1–4 days in most patients. Only 69/108 patients and 29/66 patients after the first and second dose, respectively, required treatment at home. One hundred and two children reported low-grade fever after the first dose, while 70 children developed low-grade fever after the second dose. High-grade fever was reported in 20 children after the first dose and only eight children after the second dose. Fever lasted for 4–7 days in only three children. Low-grade fever was categorized as moderate in 46/102 cases and 49/70 cases after first and second doses of the vaccine, respectively. High-grade fever was moderate on all counts, and medical consultation was sought in most cases for high-grade fever. All the patients recovered from the reported adverse events in group B (Table 5) Though one case of myocarditis was reported in group B, none of the adverse events were ‘serious’.

## 4. Discussion

Pfizer-BioNTech’s COVID-19 vaccine was the first m-RNA based vaccine to be licensed. It uses a genetically engineered mRNA, which codes for the spike protein of the virus. The mRNA is included in a lipid nano-particle and is capable of eliciting an immune response in the host. The vaccine does not contain any adjuvants or preservatives, nor is there any risk of COVID-19 from the vaccine [16]. With reports of school outbreaks of COVID-19 infection, increasing hospitalization rates and long Covid complications in children, it is important to provide safe and effective vaccines for this age group in addition to other measures to reduce transmission [17,18,19]. This cross-sectional study was conducted in order to get data from the real world and describe the incidence, severity, and result of AEFI in Indian children living in Saudi Arabia receiving the mRNA vaccine from Pfizer-BioNTech. 

Since roughly a third of the children in this study group were between the ages of 12 and 13, the average age was 12 ± 3.29 years. The proportion of men and women was equal in the study. The group of children aged 5 to 11 had a significantly higher proportion of males, however (*p* < 0.001). In the study, 92% of kids between the ages of 12 and 17 had received two doses of the vaccination at the time of the study. The immunization rate with two doses was 70% in the 5 to 11 age group since younger children could only get the vaccine a few months previous to the trial. (Table 1). 

In our study, AEFI was more commonly found after the first dose of vaccine in both the younger children and the adolescents. (*p* < 0.001) (Figure 1, Table 2) Data from patient-reported cohort event monitoring systems demonstrated a greater frequency of AEFI after the first than the second dose [20]. However, data from other studies in adults reported a greater frequency of AEFI after the second dose of vaccine [21]. Male children, especially adolescent males, had greater odds of having AEFI after the first dose of vaccine in this study (Table 2 and Table 3). In this study, there were no gender differences in the frequency of AEFI following the second dose. Earlier studies have evaluated gender differences in the frequency of AEFIs with mRNA COVID-19 vaccines and found females to have higher risk of AEFI. However, most of these studies were among adults [4,22,23,24,25,26].

Researchers found that prior COVID-19 infection was a strong predictor of AEFIs following the first and second doses of the vaccination. The study also shows fever after the first and second doses, as well as injection site responses after the second dose, were associated to prior history of COVID-19 infection. Adults with past COVID-19 infection have a greater risk of AEFI after receiving mRNA vaccinations, according to real-world vaccine safety data [25]. In this analysis, there was no correlation between AEFI and a history of allergies or adverse reactions to routine immunizations (Table 3). 

Most children reported only a single AEFI (Table 2) ‘Mild’ to ‘moderate’ injection site reaction like pain, redness, or swelling which began within 1–3 days of receiving the vaccine and subsided in 1–3 days, was most commonly reported in younger children after both the doses (Table 4, Figure 2 and Figure 3) Safety data from Phase 2/3 randomized clinical trials also reported injection site pain in 52–74% of vaccine recipients, and other local reactions as the most common AEFI in 5–11 years old children [23,24].

Study findings suggest adolescents frequently displayed multiple AEFIs, such as injection site reactions and low-grade fever following the first and second doses (Table 2, Figure 2 and Figure 3). Although, the most common AEFIs among adolescents in the United States were syncope, dizziness, and headaches, we did not receive any reports of these symptoms from the participants in this study. [4] In our study, the majority of AEFIs in adolescents started within 1 to 3 days, lasted less than 3 days, and were classified as mild to moderate. The only high fever that was reported to linger for four to seven days was treated at home with antipyretics (Table 5).

Though myalgia was commonly listed in earlier reports, only two patients in this study reported myalgia [4,5,24]. In our study, younger children frequently also reported low-grade fever combined with rhinitis.

We had one case of myocarditis, diagnosed after medical consultation, reported in a 16-year-old boy after the second dose of vaccine. The child also had reported local reactions and high-grade fever initially. Non-urgent medical consultation was sought, treatment was done at home, and the child recovered. It did not result in hospital admission. [Figure 3] The pattern of myocarditis presentation was similar to earlier studies. Previous studies have also documented increased rates of vaccine-associated myocarditis among adolescents and young adults (12–29 years) of male gender in the early days (within the 1st week to up to 21 days) after second dose of vaccination [27,28]. No cases of anaphylaxis or other AESI were reported in our study.

Once a vaccine clears the preclinical phase, it undergoes strict monitoring during the various phases of clinical trial. Only once their safety and efficacy has been established in these phases it is then approved for use in the general population. However, in spite of strict scrutiny, adverse effects related to immunization have been reported. These adverse effects are possibly due to variables such as administration pattern, immunological, genetic, and environmental factors. Influenza vaccines are commonly administered to children and adults worldwide. The Sentinel network study of eight seasons (2010–2018) reported myalgia, coughs, rashes, and headaches as common adverse effects in every age group which were observed in the first week following influenza vaccination. It is interesting to note that similar pattern of adverse effects was found with mRNA COVID-19 vaccines. 

Utilization of health resources was not common in our study population. Only non-urgent medical consultation in 4.4 to 6% children after the first and second dose was sought, mostly for fever in younger children and for local reaction plus fever in adolescents (Table 4 and Table 5) Results from other studies have also demonstrated favorable prognosis from AEFIs among children and adolescents.

Despite the fact that 13.8% of the children in this study had a confirmed COVID-19 infection after receiving two doses of the vaccine, they were all reported to be mild infections. It will not be possible to comment further on the vaccine’s efficacy because researchers lacked the necessary data to assess the children’s exposure to the virus or the seroresponse following vaccination. This is a cross-sectional study. So, accordingly we can consider the limitations of a cross-sectional study. 

## 5. Conclusions

The study showed that the m-RNA COVID-19 vaccination from Pfizer-BioNTech is safe and causes previously reported side events in Indian children between the ages of 5 and 17 that are of a mild to moderate intensity. The pattern of adverse events reporting was similar to those reported in large-scale long-term study with Influenza vaccine. After the first dose of vaccination, AEFI was more frequently found in both the younger children and the teenagers. According to this study, male children, particularly adolescent boys, were more likely to develop AEFI after receiving the first dose of the vaccine. No relationship between AEFI and a history of allergies or negative reactions to routine vaccinations was found in this investigation. 

## Figures and Tables

**Figure 1 vaccines-11-00207-f001:**
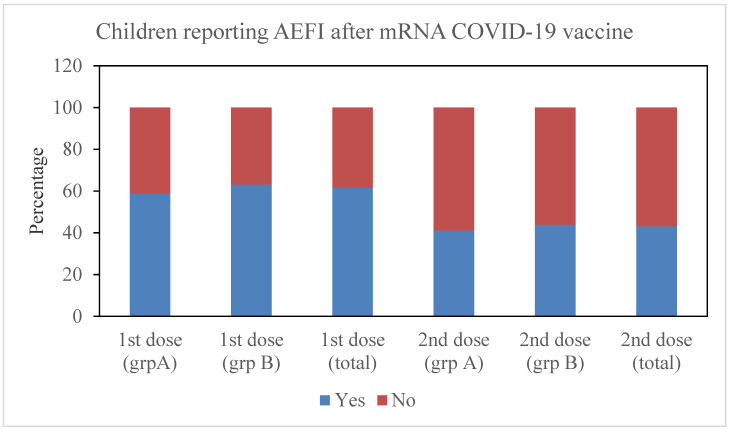
Percentage of children reporting AEFI after the first and second doses of vaccine.

**Figure 2 vaccines-11-00207-f002:**
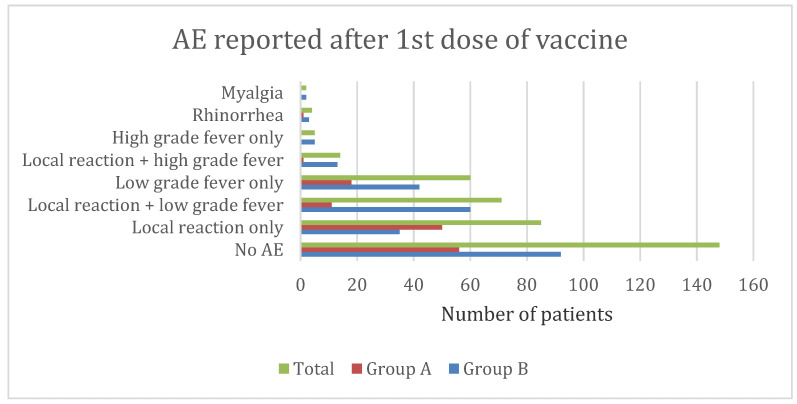
Adverse events (AE) reported following first dose of Pfizer-BioNTech’s m-RNA COVID-19 vaccine in Indian children 5–11 years of age (group A, *n* = 136) and 12–17 years of age (group B, *n* = 249). *p* < 0.001 between the groups on Chi-square test.

**Figure 3 vaccines-11-00207-f003:**
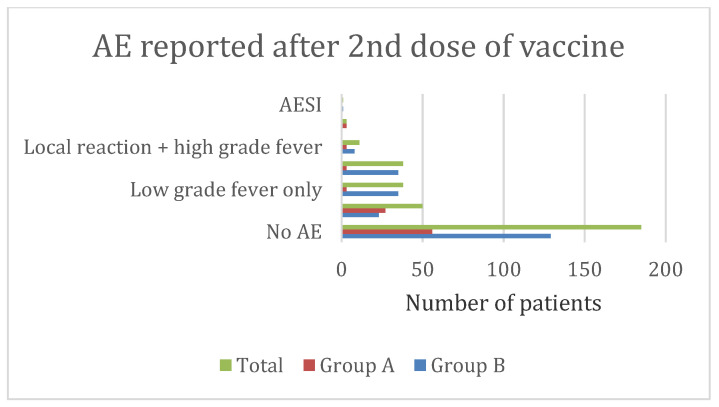
Adverse events (AE) reported following second dose of Pfizer-BioNTech’s m-RNA COVID-19 vaccine in Indian children 5–11 years of age (group A, *n* = 95) and 12–17 years of age (group B, *n* = 230). *p* < 0.001 between the groups on Chi-square test.

**Table 1 vaccines-11-00207-t001:** Demographic profile of children in group A (5–11 years) and group B (12–17 years of age) who received Pfizer-BioNTech’s mRNA-COVID-19 vaccine. The percentage is given in parentheses. $ = patients having COVID-19 infection after 2nd dose of vaccine.

Variables	Group A (*n* = 136)	Group B (*n* = 249)	Total (*n* = 385)
Region of residence			
Asir region	13 (3.3%)	6 (2.4%)	19 (5%)
Eastern region	33 (24.3%)	38 (15.3%)	71 (18.4%)
Makkah region	6 (4.4%)	23 (9.2%)	29 (7.5%)
Riyadh region	56 (41.2%)	157 (63%)	213 (55.3%)
Qasim region	18 (13.2%)	24 (9.6%)	42 (10.9%)
Others	10 (7.4%)	1 (0.4%)	11 (2.9%)
Mean Age ± SD (in years)	8.58 ± 2.06	14.26 ± 1.74	12.25 ± 3.29
Gender			
Male	84 (61.8%)	101 (40.6%)	185 (48.1%)
Female	52 (38.2%)	148 (59.4%)	200 (51.9%)
No: of doses of COVID-19 vaccine received			
One	41 (30.1%)	19 (7.6%)	60 (15.6%)
Two	95 (69.9%)	230 (92.4%)	325 (84.4%)
COVID-19 infection before vaccination			
Yes	19 (14%)	28 (11.2%)	47 (12.2%)
Maybe	7 (5.1%)	8 (3.2%)	15 (3.9%)
COVID-19 infection after vaccination	N = 95	N = 230	N = 325
Yes	11 (11.6%)	34 (14.8%)	45 (13.8%)
Maybe	3 (3.2%)	7 (3%)	10 (3.1%)
History of allergy			
Yes	15 (11%)	17 (6.8%)	32 (8.3%)
Maybe	5 (3.7%)	-	5 (1.3%)
Type of allergy			
Dust allergy	17 (12.5%)	16 (6.4%)	33 (8.6%)
Pet allergy	2 (1.5%)	-	2 (0.5%)
Allergy to antibiotics	1 (0.7%)	1 (0.4%)	2 (0.5%)
Family history of allergy			
Yes	22 (16.2%)	17 (6.8%)	39 (10.1%)
Maybe	6 (4.4%)	8 (3.2%)	14 (3.6%)
Comorbidities			
Asthma	7 (5.14%)	4 (1.6%)	11 (2.9%)
Eczema	-	1 (0.4%)	1 (0.3%)
Adrenal insufficiency	-	1 (0.4%)	1 (0.3%)
Unwanted events during past routine immunization			
Yes	3 (2.2%)	14 (5.6%)	18 (4.4%)
Maybe	-	7 (2.8%)	7 (1.8%)

**Table 2 vaccines-11-00207-t002:** Characteristics of children reporting AE after first and second dose of Pfizer-BioNTech’s m-RNA COVID-19 vaccine in group A (5–11 years) and group B (12–17 years).

Variables	Group A	Group B	Total
Total number of events reported, n			
After 1st dose	93	233	326
After 2nd dose	45	144	189
Number of patients reporting AE after 1st dose ***, n	N = 136	N = 249	N = 385
None	56 (41.2%)	92 (36.9%)	148 (38.4%)
One AE	68 (50%)	84 (33.7%)	152 (39.4%)
More than one AE	12 (8.8%)	73 (29.3%)	85 (22%)
Number of patients reporting AE after 1st dose *, n	N = 95	N = 230	N = 325
None	56 (58.9%)	129 (56.1%)	185 (56.9%)
One AE	33 (24.3%)	58 (25.2%)	91 (28%)
More than one AE	6 (4.4%)	43 (18.7%)	49 (15.1%)
Gender of patients reporting at least one AE after 1st dose, n/N ^#^			
Male	51/84 (60.7%)	75/101 (74.3%) **	126/185 (68.1%) *
Female	29/52 (55.8%)	82/148 (55.4%)	111/200 (55.5%)
Gender of patients reporting at least one AE after 2nd dose, n/N ^#^			
Male	23/62 (37.1%)	45/90 (50%)	68/152 (44.7%)
Female	16/33 (48.5%)	56/140 (40%)	72/173 (41.6%)
Cases with history of COVID-19 infection reporting at least one AE, n/N ^$^			
After 1st dose	10/19 (57.9%)	28/28 (100%)	8/47 (80.9%)
After 2nd dose ***	3/13 (23.1%)	21/23 (91.3%)	24/36 (66.7%)

AESI = adverse event of special interest. ^#^ N is the total number of male or female patients. The percentage is given in parentheses. * = *p* < 0.05, ** = *p* < 0.01, *** = *p* < 0.001 on Chi-square test. ^$^ N is the number of patients with a history of COVID-19 infection.

**Table 3 vaccines-11-00207-t003:** Description of adverse event (AE)-related variables after first and second dose of m-RNA COVID-19 vaccine in group B (12–17 years).

Predictors of AE	Odd’s Ratio	95% CI Upper Limit-Lower Limit	*p*-Value
Dose AE 1st/2nd	2.12	1.57–2.86	<0.001
Age AE after 1st dose AE after 2nd dose	0.84 0.89	0.55–1.28 0.55–1.44	0.42 0.64
Gender (Male/female) AE after 1st dose AE after 2nd dose	0.171 1.136	1.13–2.56 0.73–1.76	0.01 0.57
Prior COVID-19 AE after 1st dose AE after 2nd dose	2.95 2.98	1.38–6.3 1.44–6.2	0.05 0.003
Allergy AE after 1st dose AE after 2nd dose	0.71 0.48	0.36–1.41 0.22–1.08	0.36 0.08
AEFI with routine vaccination AE after 1st dose AE after 2nd dose	1.56 0.48	0.63–3.85 0.19–1.21	0.34 0.12

**Table 4 vaccines-11-00207-t004:** Description of adverse event (AE)-related variables after first and second dose of m-RNA COVID-19 vaccine in group A (5–11 years).

AE-Related Variables in Group A	Pain, Redness, and Swelling at the Site of Injection		Fever < 102 °F		Fever > 102 °F	
	1st Dose (*n* = 136)	2nd Dose (*n* = 95)	1st Dose (*n* = 136)	2nd Dose (*n* = 95)	1st Dose (*n* = 136)	2nd Dose (*n* = 95)
Number of events, N	62	33	29	6	1	6
Onset within, (n/N)						
1–3 days	62/62	33/33	29/29	6/6	1/1	6/6
4–7 days	-	-	-	-	-	-
Duration, (n/N)						
1–4 days	61/62	30/33	29/29	6/6	-	-
4–7 days	1/62	3/33	-	-	1/1	6/6
Intervention, (n/N)						
No treatment	45/62	24/33	14/29	-	-	-
Treatment at home	17/62	9/33	15/29	6/6	1/1	6/6
Utilization of health resources, (n/N)						
None	61/62	30/33	29/29	-	-	-
Medical consultation (non-urgent)	1/62	3/33	0/29	6/6	1/1	6/6
Outcome, %						
Recovered	100%	100%	100%	100%	100%	100%
Unknown	-	-	-	-	-	-
Severity, (n/N)						
Mild	45/62	24/33	14/29	-	-	-
Moderate	17/62	9/33	15/29	6/6	1/1	6/6
Seriousness	None	None	None	None	None	None

n = no. of events, N = total no. of events.

**Table 5 vaccines-11-00207-t005:** Description of adverse event (AE)-related variables after first and second dose of m-RNA COVID-19 vaccine in group B (12–17 years).

AE-Related Variables in Group B	Pain, Redness, and Swelling at the Site of Injection		Fever < 102 °F		Fever > 102 °F	
	1st Dose (*n* = 249)	2nd Dose (*n* = 230)	1st Dose (*n* = 249)	2nd Dose (*n* = 230)	1st Dose (*n* = 249)	2nd Dose (*n* = 230)
Number of events, N	108	66	102	70	20	8
Onset within, n/N 1–3 days 4–7 days	108/108 -	66/66 -	102/102 -	70/70 -	20/20 -	8/8 -
Duration, n/N 1–4 days 4–7 days	105/108 3/108	66/66 -	102/102 -	70/70 -	17 3	8/8 -
Intervention, n/N No treatment Treatment at home	39/108 69/108	37/66 29/66	56/102 46/102	21/70 49/70	- 20/20	- 8/8
Utilization of health resources, n/N						
None Medical consultation (non-urgent)	91/108 17/108	58/66 8/66	96/102 6/102	70/70 -	9/20 11/20	- 8/8
Outcome, % Recovered Unknown	100% -	100% -	100% -	100% -	100% -	100% -
Severity, n/N Mild Moderate	39/108 69/108	37/66 29/66	56/102 46/102	21/70 49/70	- 17/20	- 8/8
Seriousness	None	None	None	None	None	None

n = no. of events, N = total no. of events.

## Data Availability

Data will be available with from the primary author and can be made available as and when requested.

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
