# Peer review of "A Cross-Sectional Study to Assess mRNA-COVID-19 Vaccine Safety among Indian Children (5–17 Years) Living in Saudi Arabia"

_vaccines, 2023, doi:10.3390/vaccines11020207_

Round 1
Reviewer 1 Report
The paper should be improved before publication.
The introduction is too generic and should include aspects that introduce the reader to more specific aspects of the study such as the population of Indian children living in Saudi Arabia.
The objectives of the study should be clearly specified at the end of the introduction section.
Regarding the methodology used to collect the information in the study, the authors should explain the procedure used. In any scientific study the methodology is of great importance to know the representativeness of the sample and the extrapolation of the results to the target population.
According to the authors, the information was collected from parents and guardians. However, it would be interesting to expand the information on this aspect. It is necessary to take into account the educational level of the individuals who answered the questionnaire and that there are subjects who did not know how to adequately describe the adverse effects that the vaccine caused in their children. This issue introduces a bias in the study that should be mentioned. Nor do the authors report the number of rejected questionnaires.
The results section is well described and explained. However, the discussion section is poor and is practically a reiteration of the results previously presented.
The conclusions section should also be improved.
Author Response
Point 1: The introduction is too generic and should include aspects that introduce the reader to more specific aspects of the study such as the population of Indian children living in Saudi Arabia.
Response-1: Thank you very much for the comment. As per the Government of India data, Indian population in Saudi Arabia is the largest expatriate population which stands at 2.5 million. There is no data which mentions the Indian children population. But considering the sizable Indian population and also with 39 Indian Schools (as per the Embassy of India, Riyadh) in the Kingdom, each having an average of 5000 students, with schools in major cities having more than 10000 students, we can approximately calculate it to be 0.2 million. We have modified the introduction section of the manuscript.
Point 2: The objectives of the study should be clearly specified at the end of the introduction section
Response-2: Thank you very much for the comment. We have added the aim at the end of the introduction
Point 3: Regarding the methodology used to collect the information in the study, the authors should explain the procedure used. In any scientific study the methodology is of great importance to know the representativeness of the sample and the extrapolation of the results to the target population.
Response-3: Thank you for the comment. The data was collected through google form which was distributed among the Indian population using snowball sampling technique through Indian schools. The sampling technique has been added in the methodology section.
Point 4: According to the authors, the information was collected from parents and guardians. However, it would be interesting to expand the information on this aspect. It is necessary to take into account the educational level of the individuals who answered the questionnaire and that there are subjects who did not know how to adequately describe the adverse effects that the vaccine caused in their children. This issue introduces a bias in the study that should be mentioned. Nor do the authors report the number of rejected questionnaires.
Response-4: Thank you for your comment. The questionnaires distributed to the parents contained basic and relevant information pertaining to adverse effects related to the vaccines and its reporting. This was done to ensure that the participant got all the necessary information regarding adverse effects without getting confused with the complex medical terminologies. The details are added in the methodology section. As we used snowball technique for data collection, we cannot determine how many participants received the questionnaires. Hence, we could not calculate the response rate. We kept the Google form link open till sample size fulfilling the inclusion criteria was achieved. 408 responses were received, of which 23 did not fulfill the inclusion criteria for various reasons. This has been added to the methodology section.
Point 5: The results section is well described and explained. However, the discussion section is poor and is practically a reiteration of the results previously presented.
Response-5: Thank you for your appreciation. We have modified the discussion section.
Point 6 The conclusions section should also be improved.
Response-6: Thank you for the comments. We have modified the conclusion.

Reviewer 2 Report
This paper reports on adverse events and severity following Covid-19 vaccination of Indian children living in Saudi Arabia. Although it contains data that may be useful to the reader, I recommend that this paper be accepted after revisions and additions are made to the four points listed below.
1: Regarding the subject matter.
Why is there a need to even indicate in the title the results of a study that vaccinated Indian children in Saudi Arabia? As the authors state, the fact that immunological responses among ethnic groups are varied is well understood. However, from a different point of view, it could be taken as discrimination, so why not mention it carefully?
2: The arrangement of the Indexes in the Table 1 table is not aligned, making it difficult to understand.
We hope this will be improved.
3: Figures 2 and 3 show AEs after 1st and 2nd vaccination. Hence the reader is supposed to compare and contrast the readings. Therefore, it would be better to unify the scale of No. patients shown on the horizontal axis and make it one Figure.
4: In the Abstract, the Conclusions states that Covid-19 vaccination was found to be safe for children. However, at the end of the Discussion, it states that 13.8% of children were infected after the second dose of vaccination. What the authors need to clarify about these two points is the definition of vaccine. It should be clearly stated whether the vaccine effect is to control infection or to control the severity and incidence of the disease.
Author Response
Point 1: Why is there a need to even indicate in the title the results of a study that vaccinated Indian children in Saudi Arabia? As the authors state, the fact that immunological responses among ethnic groups are varied is well understood. However, from a different point of view, it could be taken as discrimination, so why not mention it carefully?
Response-1: Thank you for your comments. The study was done exclusively in Indian children to gather data from this ethnic groups. As the study was conducted in Saudi Arabia, it has been mentioned in the title.
Point 2: The arrangement of the Indexes in the Table 1 table is not aligned, making it difficult to understand. We hope this will be improved.
Response-2: Thank you for your response. We acknowledge the issue with table 1. The arrangement of the indices and designing of the table was done by the editorial office. However, we have made some modification in the table to make it easier to understand.
Point 3: Figures 2 and 3 show AEs after 1st and 2nd vaccination. Hence the reader is supposed to compare and contrast the readings. Therefore, it would be better to unify the scale of No. patients shown on the horizontal axis and make it one Figure.
Response -3: Thank you for your advice and comments. A detailed comparison of common adverse effects in both groups after the 1st and 2nd dose of vaccination is mentioned in table 4 and table 5.
Point 4: In the Abstract, the Conclusions states that Covid-19 vaccination was found to be safe for children. However, at the end of the Discussion, it states that 13.8% of children were infected after the second dose of vaccination. What the authors need to clarify about these two points is the definition of vaccine. It should be clearly stated whether the vaccine effect is to control infection or to control the severity and incidence of the disease.
Response-4: Thank you for your valuable advise and comment. Though there were cases of Covid-19 after the vaccination, the parents stated that these symptoms were mild. We have highlighted the same in the discussion.

Reviewer 3 Report
A very important publication. The data make it possible to objectify information on the safety of vaccinations - not only against COVID-19.
Valuable work, material properly described, results clear, well discussed.
In order to improve the quality of work, please consider a few points:
- has swelling/redness been assessed by a doctor/nurse? So was the assessment objective? What diameters were considered significant?
- what was the accepted classification of pain at the puncture site - mild/moderate - it is not clearly defined
- relatively many children diagnosed with asthma, what is the reason for this? location? climate? typical values for this population? Or maybe overdiagnosis of asthma?
- it would be good to compare them to classic vaccines and cite several works on the safety of vaccinations (e.g. flu vaccine in chronically ill people, engerix, twinrix - immunological answer, economic prerequisites of active influenza prevention). The purpose of such a procedure would be to emphasize that in the case of "recognised" vaccinations, the safety was already assessed many, even 20 years ago, and vaccinations against COVID-19 are not less safe in this situation. The latter remark is important due to the social impact of the publication.
Author Response
Point 1: A very important publication. The data make it possible to objectify information on the safety of vaccinations not only against COVID-19. Valuable work, material properly described, results clear, well discussed.
Response-1: We sincerely thank you for appreciating our work. We hope to act on your valuable suggestions in order to further improve the quality of our paper.
Point 2: Has swelling/redness been assessed by a doctor/nurse? So was the assessment objective? What diameters were considered significant?
Response-2: Thank you for your comments. As per the FDA, reporting of adverse reaction can be done by healthcare provider or reported voluntarily by patients or attendants through various portals. During Covid-19 due to limited direct access to hospitals, most of adverse effects were reported voluntarily on mobile apps, SFDA website or through telemedicine portal set up the Ministry of Health, Kingdom of Saudi Arabia. Moreover, in order to sensitize the parents to adverse effects and increase their awareness, the questionnaire included information regarding Covid-19 vaccination, AEFI and reporting of AEFI.
Point 3: What was the accepted classification of pain at the puncture site - mild/moderate - it is not clearly defined
Response-3: Thank you for your comments. We used Hartwig-Seigel Severity scale for assessment of severity of any adverse event.
Point 4: Relatively many children diagnosed with asthma, what is the reason for this? location? climate? typical values for this population? Or maybe overdiagnosis of asthma?
Response-4: Thank you for your comments. In a report by Saudi Initiative for Asthma – 2021, it was reported that the prevalence of asthma in children ranges between 8% to 25% depending upon the location within the kingdom with lower prevalence in higher altitudes and higher prevalence in desert areas particularly in the last three decades. The reason for relatively higher incidence of asthma in children may be attributed to various reasons such as rapid lifestyle changes in view of the modernization, exposure to environmental factors such as allergens, dust, sandstorms etc.
Point 5: it would be good to compare them to classic vaccines and cite several works on the safety of vaccinations (e.g. flu vaccine in chronically ill people, engerix, twinrix - immunological answer, economic prerequisites of active influenza prevention). The purpose of such a procedure would be to emphasize that in the case of "recognised" vaccinations, the safety was already assessed many, even 20 years ago, and vaccinations against COVID-19 are not less safe in this situation. The latter remark is important due to the social impact of the publication.
Response-5: Thank you for your valuable advice. We have added data from The Sentinel network study of eight seasons (2010–2018) done on influenza vaccine and compared the observations of both studies.

Round 2
Reviewer 1 Report
The paper has been improved according to the suggestions.
It can be published if the Editorial Committee deems it appropriate.
Reviewer 3 Report
after introducing corrections, the paper may be considered for publication